

# Fumaric acid production from fermented oil palm empty fruit bunches using fungal isolate K20: a comparison between free and immobilized cells

Antika Boondaeng[1], Jureeporn Keabpimai[1], Chanaporn Trakunjae[1] and Nanthavut Niyomvong[2,3]

[1] Kasetsart Agricultural and Agro-Industrial Product Improvement Institute, Kasetsart University, Bangkok, Thailand
[2] Department of Biology and Biotechnology, Faculty of Science and Technology, Nakhon Sawan Rajabhat University, Nakhonsawan, Thailand
[3] Science Center, Nakhon Sawan Rajabhat University, Nakhonsawan, Thailand

## ABSTRACT

This study investigated the potential of using steam-exploded oil palm empty fruit bunches (EFB) as a renewable feedstock for producing fumaric acid (FA), a food additive widely used for flavor and preservation, through a separate hydrolysis and fermentation process using the fungal isolate K20. The efficiency of FA production by free and immobilized cells was compared. The maximum FA concentration (3.25 g/L), with 0.034 g/L/h productivity, was observed after incubation with the free cells for 96 h. Furthermore, the production was scaled up in a 3-L air-lift fermenter using oil palm EFB-derived glucose as the substrate. The FA concentration, yield, and productivity from 100 g/L initial oil palm EFB-derived glucose were 44 g/L, 0.39 g/g, and 0.41 g/L/h, respectively. The potential for scaling up the fermentation process indicates favorable results, which could have significant implications for industrial applications.

## INTRODUCTION

Fumaric acid (FA) is a four-carbon unsaturated dicarboxylic acid with various applications in multiple industries, including the chemical, pharmaceutical, cosmetic, and food industries. Traditionally, it is used as a food additive, acidulant in foods and beverages, livestock feed supplement, and an intermediate for polymer production (*Roa Engel et al., 2008*). However, its commercial production typically involves a chemical synthesis process using nonrenewable resource-derived maleic anhydrides, such as butane (*Roa Engel et al., 2008*), which can have negative environmental impacts, particularly for global warming.

In recent years, there has been a growing emphasis on shifting from petroleum-based resources for chemicals, materials, and liquid fuels to more sustainable alternatives. Renewable resources have become the focal point in the development of sustainable energy. Research conducted at the National Renewable Energy Laboratory (NREL) in

Corresponding author
Nanthavut Niyomvong,
nanthavut.ni@nsru.ac.th

Colorado, USA, has highlighted the potential of chemical compounds derived from lignocellulosic biomass as crucial building blocks for producing valuable chemical substances (*Werpy & Petersen, 2004*). Therefore, new methods for producing FA for use in food and pharmaceutical industries are necessary. FA production through bioconversion of renewable resources has recently gained significant attention (*Goldberg, Rokem & Pines, 2006*). *Gangl, Weigang & Keller (1990)* reported that the cost of raw materials is a significant obstacle to industrial FA production. Using refined sugars such as glucose and sucrose as feedstock for FA production is costly. It is more economical to use cheaper, abundant, sustainable, and readily available feedstock.

Conventional FA production from lignocellulosic materials requires pre-treatment of materials for separate hydrolysis and fermentation (SHF) operations, as outlined by *Carta et al. (1999)* and *Moresi et al. (1992)*. The SHF process consists of two distinct stages, saccharification and fermentation, with independent reactors. In this process, the pre-treated lignocellulosic biomass is first enzymatically saccharified at the optimum temperature for enzymatic hydrolysis, followed by adding microorganisms to ferment the resulting hydrolysate. Several microbial species, including *Rhizopus*, *Mucor*, *Cunninghamella*, and *Circinella* spp., produce FA, with the *Rhizopus* species being the most efficient (*Zhou, Du & Tsao, 2002*; *Liao, Liu & Chen, 2007*). Previous studies have reported FA production using dairy manure, starchy materials, and glycerol (*Liao, Liu & Chen, 2007*; *Deng et al., 2012*; *Li et al., 2014*). However, the use of oil palm empty fruit bunches (EFB) for FA production remains limited. Oil palm EFB, an agricultural residue abundantly available worldwide after palm oil extraction, is widely used in ethanol or aromatic compound production, lignin preparation, and other processes (*Jeon et al., 2014*; *Lee et al., 2015*; *Medina et al., 2015*). It contains approximately 34.6% cellulose and 17.1% hemicellulose (*Jeon et al., 2014*), making it a promising resource for FA production. The utilization of lignocellulosic biomass, such as oil palm EFBs, is an attractive and environment-friendly alternative for FA production. By exploring the potential of fungal strains isolated from natural resources, researchers can optimize the production process and contribute to the sustainable utilization of renewable resources. This research direction aligns with the ongoing efforts to create a sustainable and environmentally conscious chemical industry.

This study was conducted to screen and isolate organic acid-producing fungi, investigate the possibility of FA production from oil palm EFB through a fermentation process using a fungal strain, and evaluate the optimal conditions for FA production using statistical analysis.

## MATERIALS AND METHODS

### Fungi isolation

A total of 26 soil samples were collected from Bangkok and Nakhon Ratchasima provinces in Thailand separately in plastic bags, taken to a laboratory at Kasetsart University in Bangkok, stored at 4 °C, and analyzed within 24 h of collection. Field experiments were conducted under the principles and ethical standards approved by the National Research

Council of Thailand (NRCT) and Thailand Science Research and Innovation (TSRI), (Approval reference: MHESI 6390.FB. 6.1/1/2564).

One gram soil sample was suspended in sterile 0.85% saline solution to prepare serial dilutions spread in triplicate on potato dextrose agar (PDA) plates. The plates were then incubated at 30 °C for 48–168 h. The fungal isolates were selected based on morphological characteristics, sub-cultured using PDA slants, and stored at 4 °C.

## Screening for organic acid-producing microorganisms

Fungal isolates were inoculated on a mineral agar acid indicator medium containing 120 g/L glucose, 3.02 g/L $(NH_4)_2SO_4$, 0.25 g/L $MgSO_4 \cdot 7H_2O$, 0.04 g/L $ZnSO_4 \cdot 7H_2O$, 0.15 g/L $KH_2PO_4$, 20 g/L agar, 0.2 g/L bromocresol green, and 1.5 mL/L Triton X-100 in distilled water (pH: 5.5; *Suntornsuk & Hang, 1994*). Fungal isolates that form a yellow zone around the colony, indicating their ability to produce acid, were selected and tested for FA production.

## Screening for FA production

The fungal isolates were cultured on PDA plates at 30 °C for 7 days. The spores were then washed twice with sterile water to obtain a spore suspension. After filtration through sterile cotton layers, the spore suspension was counted under a microscope using a hemocytometer (Boeco Germany, Hamburg, Germany) and diluted to the desired concentration. For the seed culture, 1 mL spore suspended in distilled water ($1 \times 10^7$ spores/mL) was inoculated into the growth medium containing 30 g/L glucose, 0.6 g/L $KH_2PO_4$, 0.0176 g/L $ZnSO_4 \cdot 7H_2O$, 1.5 g/L urea, 0.5 g/L $MgSO_4 \cdot 7H_2O$, 0.0005 g/L $FeSO_4 \cdot 7H_2O$ (pH: 7). After incubating for 24 h, the cell pellet was washed with autoclaved distilled water and fermented in the production medium containing 90 g/L glucose, 0.6 g/L $KH_2PO_4$, 0.04 g/L $ZnSO_4 \cdot 7H_2O$, 0.05 g/L urea, 0.5 g/L $MgSO_4 \cdot 7H_2O$, 1.25 g/L $(NH_4)_2SO_4$, 0.1 g/L $Na_2CO_3$ (pH: 5), in a static condition at 30 °C for 168 h. The samples were aliquoted every 24 h for determining the FA concentration. The experiment was done in triplicate.

## Fungal isolate identification

Genomic DNA was extracted from the mycelia using an E.Z.N.A. Forensic DNA kit (Omega Bio-tek Inc., Norcross, GA, USA), following the manufacturer's instructions. The DNA purified using this kit was suitable for polymerase chain reaction (PCR) analysis.

The internal transcribed spacer (ITS) region was amplified in a 50-mL reaction mixture (Applied Biosystems, Foster City, CA, USA) containing 1 U Taq DNA polymerase, 1 × buffer, 2.5 mM $MgCl_2$, 0.2 mM dNTPs, and 0.2 μM each primer (ITS5 and ITS4). All the primers were synthesized by Integrated DNA Technologies (Coralville, IA, USA). The PCR conditions were pre-denaturation at 96 °C for 2 min; 35 cycles of denaturation at 96 °C for 1 min, annealing at 53 °C for 1 min, and extension at 72 °C for 1.5 min; final extension at 72 °C for 10 min. The purified amplified genomic DNA was electrophoresed in ethidium bromide-stained 1% agarose (Fluka Biochemika, Gillingham, UK) and observed under ultraviolet light. The amplified and purified genes were sequenced in both directions using

an automated DNA sequencer (Macrogen Inc., Seoul, Korea). Nucleotide sequences obtained from all primers were assembled using the Cap Contig Assembly Program v.3 (*Huang & Madan, 1999*), BioEdit Program v.7.2.5 (a biological sequence alignment editor program; *Hall, 1999*). The sequences were compared with nucleotide sequences in GenBank, CBS, or a suitable database.

## Raw material and reference microorganisms

This study used oil palm EFB sourced from the Suksomboon Palm Oil Industry (Chonburi Province, Thailand) as the raw material. The EFB were sun-dried, ground to obtain approximately 2.5 cm particles, and then stored in a sealed plastic bag until further use.

Two commercial strains, *R. oryzae* TISTR 3535 and *R. arrhizus* TISTR 3198 (NRRL 1469), were procured from the Thailand Institute of Scientific and Technological Research (TISTR) and used as reference strains. These strains were cultured and maintained on PDA agar at 4 °C until they were ready for use.

## Raw material pre-treatment

Approximately 200 g oil palm EFB was subjected to steam explosion at 20 MPa (210 °C) for 4 min in a 2.5-L stainless steel batch digester (Nitto Koatsu Co. Ltd., Tokyo, Japan) (adapted from *Siramon, Punsuvon & Vaithanomsat, 2018*). The material was then separated into a solid residue and liquid by filtration through a cheesecloth. The solid residue was soaked in hot water at 80 °C for 60 min, washed with tap water until it reached a neutral pH, and further delignified.

The steam explosion-pretreated materials were delignified by soaking in 15% (w/v) NaOH and incubated at 90 °C for 30 min. Subsequently, the suspension was filtered and the solid residue was washed with tap water until it reached a neutral pH. The delignified solid was air-dried and stored in a sealed plastic bag for further separate hydrolysis and fermentation (SHF).

## Separate hydrolysis and fermentation (SHF)

The steam-exploded and delignified oil palm EFB were individually used for FA production *via* SHF. The raw material was enzymatically saccharified using a Cellic CTec 2 (185FPU/mL; Novozyme A/S, Basgsværd, Denmark) in a citrate buffer (50 mM, pH 4.8) at 50 °C and 150 rpm for 24 h. The enzyme loading was fixed at 15 FPU/g of raw material. The glucose concentration was quantitatively determined using a high-performance liquid chromatography column (HPLC, Shimadzu LC-20A; Shimadzu, Kyoto, Japan). Briefly, 100 mL enzymatic hydrolysate with initial oil palm EFB-derived glucose of 3.5% was used for FA fermentation in a 250-mL Erlenmeyer flask supplemented with the seed culture and production medium, except that the enzymatic hydrolysate was used instead of glucose. Static fermentation was performed at 30 °C for 168 h. The samples were aliquoted every 24 h for determining the FA concentration.

## Comparison of immobilized and free cells

To compare the efficiency of free and immobilized cells for FA production, the fungal isolate was prepared in either free or immobilized form. For the free cells, 1% spores (1 ×

$10^7$ spores/mL) were inoculated to a 250-mL flask containing 100 mL growth medium, and then incubated at 30 °C and 150 rpm for 24 h. The cells were then harvested, washed twice with sterile distilled water, and transferred into production medium. For the immobilized cells, a 0.5 mm × 0.5 mm × 0.5 mm polyurethane sponge was used as a support for the cell immobilization process. The polyurethane cube was placed in a 250-mL Erlenmeyer flask containing 100 mL growth medium and inoculated with the fungal spore suspension. After incubation for 24 h in a rotary shaker at 30 °C with 150 rpm, the growth medium was removed. The immobilized cells were washed twice with sterile water, transferred to the production medium, and cultured at 30 °C and 150 rpm in the rotary shaker for 96 h. The samples were aliquoted every 24 h to determine the FA concentration.

### Batch fermentation in an air-lift fermenter

Batch fermentation was conducted in a 3-L air-lift fermenter (MCI-6C; B.E. Marubishi Co. Ltd., Bangkok, Thailand) with 2 L working volume with 80, 100, and 120 g/L oil palm EFB-derived glucose instead of pure refined glucose as the substrate. The fermentation condition was fixed at 30 °C, initial pH of 3.5, and 1.5 volume of gas per volume of liquid per minute (VVM) aeration rate (*Boondaeng et al., 2022*). The samples were aliquoted every 24 for 120 h to determine the FA concentration using high-performance liquid chromatography (HPLC).

### Analytical methods

An oil palm EFB sub-sample was milled using a Wiley mill (Kinematica AG Co., Ltd., Tokyo, Japan) to 40-mesh particles. Its chemical composition was quantitatively determined according to the TAPPI standard methods: *TAPPI T204 om-88 (1997)* for extractives; *TAPPI T211 om-93 (2002)* for ash; *TAPPI T222 om-88 (1988)* for Klason (acid-insoluble) lignin; *TAPPI T223 cm-84 (2001)* for pentosan; and *TAPPI T203 om-88 (1992)* for α-cellulose. An HPLC with an Aminex HPX-87H column (300 × 7.8 mm²; Bio-Rad, Richmond, USA) was used to measure FA concentration quantitatively. The mobile phase was 0.005 $NH_2SO_4$ at 40 °C and 0.6 mL/min flow rate.

The glucose concentration was quantitatively determined a high-performance liquid chromatography column (HPLC, Shimadzu LC-20A; Shimadzu, Kyoto, Japan) consisting of a refractive index detector (RID). The analyses were separated on an Aminex HPX-87P column at 80 °C. The HPLC system operated at a flow rate 0.6 mL/min in deionized water as the mobile phase (*Liu et al., 2010*).

## RESULTS

### Fungal sample isolation and identification

In total, 463 fungal isolates were obtained from 26 soil samples, of which 15 isolates significantly produced FA (Table 1). Isolate K20 produced the maximum FA (3.25 g/L) after 96 h of cultivation, while the reference strains *R. oryzae* TISTR 3535 and *R. arrhizus* TISTR 3198 produced 0.12 and 0.09 g/L FA, respectively. After confirming its ability to produce FA, isolate K20 was selected for further investigation. The morphological characteristics of the isolate K20 are shown in Fig. 1A. Initially, an isolate K20 colony

**Table 1 Table of fumaric acid yield of fungal isolates.**

| Fungal isolate | Fumaric acid concentration (g/L) |
| --- | --- |
| N65 | 0.1 ± 0.00 |
| K9 | 0.15 ± 0.03 |
| K12 | 0.11 ± 0.01 |
| K13 | 0.05 ± 0.00 |
| K20 | 3.25 ± 0.01 |
| K25 | 0.35 ± 0.04 |
| K53 | 0.14 ± 0.00 |
| K55 | 1.07 ± 0.01 |
| K80 | 0.48 ± 0.08 |
| K120 | 0.55 ± 0.01 |
| K121 | 0.16 ± 0.03 |
| K130 | 0.18 ± 0.00 |
| K147 | 0.04 ± 0.00 |
| K152 | 0.1 ± 0.00 |
| K189 | 0.47 ± 0.01 |
| TISTR 3535 | 0.12 ± 0.01 |
| TISTR 3198 | 0.09 ± 0.00 |

**Note:**
Fumaric acid yield of fungal isolates

growing on PDA appeared white and cottony. Over time, it became heavily speckled with sporangia, gradually turning blackish-gray, and spreading rapidly to the substrate with stolons through the rhizoids (Fig. 1A). The globose to ovoid single-celled sporangiophores, originating from the rhizoids within a spherical structure known as the sporangium, are straight, smooth-walled, and responsible for producing numerous multinucleated spores (Fig. 1B). Based on its morphological and molecular characteristics, isolate K20 closely resembled *R. arrhizus*. The identity of isolate K20 was confirmed by amplifying and sequencing the ITS region using ITS1. The isolate was placed within a clade comprising *R. arrhizus* reference isolates (90.31%). The phylogenetic tree constructed using the NJ method showed that isolate K20 was closely related to the subclade *R. arrhizus* (Fig. 2). Isolate K20 was 90.31% similar to *R. arrhizus*. Based on the morphological characteristics and the phylogenetic tree, isolate K20 exhibits characteristics that suggest that it may warrant classification as a new genus, a matter that requires further investigation. The ITS gene sequence of this isolate has been deposited in the GenBank database under the accession number OR717492.

## Chemical composition of oil palm EFB

The chemical compositions of the oil palm EFB before and after steam explosion pre-treatment are listed in Table 2. The cellulose content of oil palm EFB before and after steam explosion pre-treatment was 65.48% and 75.25%, respectively. In contrast, the percentages of hemicellulose, lignin, and extractive in ethanol/benzene decreased from 22.19%, 15.18%, and 3.21% to 2.02%, 7.50%, and 2.11%, respectively, after the steam

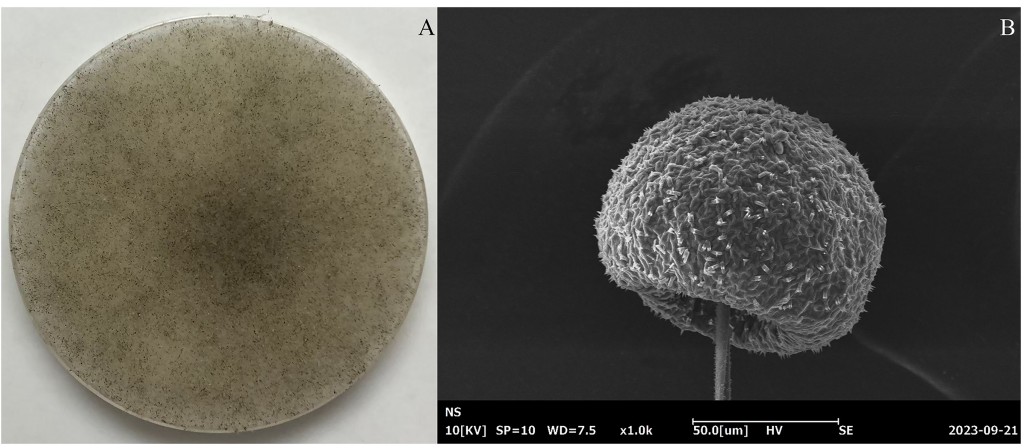

**Figure 1** (A) Morphological and (B) scanning electron microscopical features of fungal isolate K20 grown on a potato dextrose agar (PDA) plate.

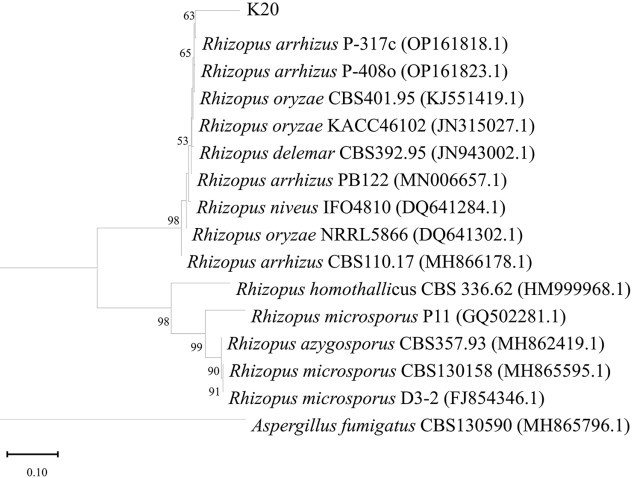

**Figure 2** Neighbor-joining phylogenetic tree generated from maximum likelihood analysis based on ITS rDNA sequences of isolate K20 and related *Rhizopus* species. Bootstrap values above or equal to 50% (1,000 replicates) are shown at each branch. Outgroup: [i]As.

**Table 2** Table of chemical composition.

| Component | Before pre-treatment | After pre-treatment |
|---|---|---|
| Cellulose | 65.48 ± 0.53 | 75.25 ± 0.21 |
| Hemicellulose | 22.19 ± 0.37 | 2.02 ± 0.05 |
| Lignin | 15.18 ± 0.7 | 7.50 ± 0.42 |
| Extractive in ethanol/benzene | 3.21 ± 0.24 | 2.11 ± 0.48 |

**Note:**
Chemical composition (% dry weight) of oil palm empty fruit bunch (EFB) before and after steam explosion pre-treatment. Data are presented as mean ± SE.

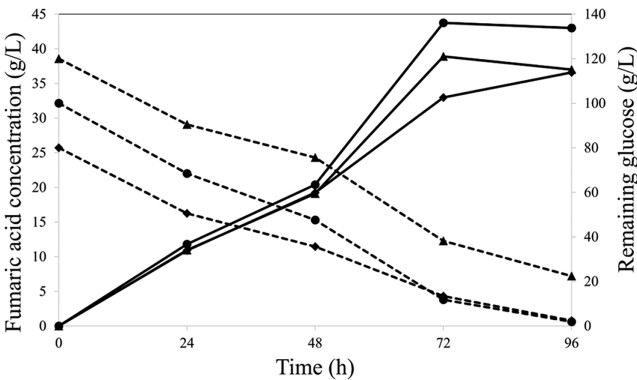

**Figure 3 Fumaric acid (solid line) and remaining glucose (dash line) concentrations at initial glucose concentrations (g/L) of 80 (♦), 100 (●), and 120 (▲).**

explosion pre-treatment. The cellulose content of other materials, such as *Acacia mangium*, *Acacia hybrid* (*Boondaeng et al., 2015*), sugarcane bagasse (*Guilherme et al., 2015*), and corncobs (*Liu et al., 2010*) is 44%, 42%, 38.6%, and 38.8%, respectively, before pre-treatment. The cellulose content of pre-treated *A. mangium*, *A. hybrid*, sugarcane bagasse, and corncobs was 76%, 74%, 65%, and 65.7%, respectively.

## Comparison of immobilized and free cells

Immobilization techniques have been widely explored to increase the possibility of a continuous fermentation process, as well as reuse microbial cells for several cycles on an industrial scale (*Naude & Nicol, 2017*; *Deng & Aita, 2018*; *Swart, Brink & Nicol, 2022*). The amount of FA produced after fermentation for 96 h with free and immobilized isolate K20 cells was 3.25 and 1.50 g/L, respectively. Thus, free cell fermentation had a higher volumetric productivity (0.034 g/L/h) than immobilized cell fermentation (0.016 g/L/h).

## Batch fermentation in an air-lift fermenter

A 3-L air-lift fermentation was used to upscale FA production. Oil palm EFB-derived glucose was used instead of commercial glucose at the initial concentrations of 80, 100, and 120 g/L. The highest FA concentration (44 g/L) was obtained within 72 h of fermentation with 100 g/L initial oil palm EFB-derived glucose (Fig. 3). The yield and productivity were 0.44 g/g and 0.61 g/L/h, respectively, which were 13.54-fold higher than those in a volumetric flask. These results indicate that isolate K20 can produce FA from lignocellulosic material-derived glucose hydrolysate.

## DISCUSSION

The highest FA concentration obtained from free-cell fermentation is consistent with the results obtained by *Deng et al. (2012)*, who demonstrated higher FA concentration and productivity from free-cell fermentation than those from immobilized cell fermentation with mutant *R. oryzae* DG-3 strain. However, some studies have reported different patterns in FA concentration and productivity for free and immobilized cells. *Gu et al. (2013)* reported the optimal performance for fermentation with free and immobilized

*R. arrhizus* RH-07-13 cells in a shake flask. Although the final FA concentration for fermentation with free and immobilized cells was slightly different (31.23 and 32.03 g/L, respectively), the volumetric productivity for fermentation with immobilized cells (1.335 g/L/h) was six times higher than that for fermentation with free cells (0.217 g/L/h). *Ronoh et al. (2022)* reported the effect of CaCO3 on the co-production of FA and malic acid by the immobilized *R. delemar*. The increased concentration of calcium ions to 10 g/L resulted in a three-fold enhancement of MA titres. Additionally, the raising of pH above 7 was correlated with a decrease in the concentration of FA. At pH 4.0, FA was produced at 10.1 g/L and then hydrated to 7.41 g/L of malic acid at pH 7.0. Moreover, *Swart et al. (2022)* studied FA production by immobilized *R. oryzae* with continuous fermentation. It was found that medium pH and the urea feed rate affected FA production. At pH 4.0, a urea feed rate of 0.625 mg/L/h and a glucose feed rate of 0.329 g/L/h yielded 0.93 g/g of FA. Thus, fungal cell immobilization for fermentation is complex process influenced by various parameters. One crucial parameter is fungal morphology, which is directly affected by culture conditions, including agitation rate, temperature, medium composition, initial spore concentration, and the immobilization matrix used (*Ronoh et al., 2022*). Our study demonstrated that, despite the potential use of immobilized cells for FA fermentation, they proved to be less efficient and productive compared to free cells. This could be attributed to the immobilization process somehow diminishing the fermentative capacity of isolate K20 for fumaric acid production.

The oil palm EFB was utilized as a substrate for FA fermentation in this study. The cellulose content reported in this study is comparable to that of both *Acacia* species, sugarcane bagasse, and corncob (75.25% *vs.*, 74–76%, 65%, and 65.7%); however, oil palm EFB is a waste of the oil palm industry and abundantly available in Thailand. Therefore, it can be a renewable resource for FA production. Lignocellulose is the most abundant renewable resource and an inexpensive feedstock for industrial fermentation. Few studies have reported FA production from lignocellulosic materials. *Li et al. (2017)* reported a higher FA production of 41.32 g/L, with a substrate yield of 0.21 g/g, from alkali-pretreated corncob in the combined and fed-batch SSF processes. The highest yield of 0.54 mg/g was achieved when wheat bran was used as substrate through simultaneous saccharification and fermentation (*Jiménez-Quero et al., 2017*) by *A. oryzae*. *Abraham et al. (2020)* studied FA production from sugarcane trash hydrolysate using *R. oryzae* NIIST1. The highest FA production in the medium containing sugarcane trash hydrolysate was 5.2 g/L. *Swart et al. (2022)* reported that FA production can be produced from a synthetic lignocellulosic hydrolysate (glucose–xylose mixture) by *R. oryzae* ATCC 20344. FA yield of 0.735 g/g was achieved at a specific substrate feed rate of 0.164 g/L/h in batch and continuous fermentations. In this research, after the fermentation process was scaled up in the 3-L air-lift fermenter, the FA concentration was 13.54 fold higher than that in the volumetric flask. The results show that upscaling FA production induces a higher concentration and improves fermentation productivity and yield.

## CONCLUSIONS

In this study, organic acid-producing fungi were successfully isolated from natural soil samples. Isolate K20 has been described as a potential FA producer. Fermentation with the free cells yielded 3.25 g/L FA compared with the 1.50 g/L FA by fermentation with the immobilized cells. An air-lift bioreactor system enhances FA production by isolate K20 using oil palm EFB-derived glucose as the substrate. Consequently, it can be concluded that oil palm EFB could be a potential renewable resource for FA production. The FA concentration and productivity were improved by scaling up the air-lift fermenter.

## ACKNOWLEDGEMENTS

The authors thank the Kasetsart Agricultural and Agro-Industrial Product Improvement Institute (KAPI; Bangkok, Thailand) for providing facilities and various tools necessary for the study.

### Funding

This work was supported by the Kasetsart University Research and Development Institute (KURDI; Bangkok, Thailand) under the New Generation of Researchers Fund (KURDI 25.57). ARDA provided research funding for this work. The funders had no role in study design, data collection and analysis, decision to publish, or preparation of the manuscript.

### Grant Disclosures

The following grant information was disclosed by the authors:
Kasetsart University Research and Development Institute (KURDI) under the New Generation of Researchers Fund: KURDI 25.57.
ARDA.

### Competing Interests

The authors declare that they have no competing interests.

### Author Contributions

- Antika Boondaeng conceived and designed the experiments, performed the experiments, prepared figures and/or tables, authored or reviewed drafts of the article, and approved the final draft.
- Jureeporn Keabpimai performed the experiments, analyzed the data, prepared figures and/or tables, and approved the final draft.
- Chanaporn Trakunjae performed the experiments, analyzed the data, prepared figures and/or tables, and approved the final draft.
- Nanthavut Niyomvong conceived and designed the experiments, performed the experiments, analyzed the data, authored or reviewed drafts of the article, and approved the final draft.
## Field Study Permissions

The following information was supplied relating to field study approvals (*i.e.*, approving body and any reference numbers):

Field experiments were approved by National Research Council of Thailand (NRCT).

## Data Availability

The raw data are available in the Supplemental Files.

## Supplemental Information

Supplemental information for this article can be found online at http://dx.doi.org/10.7717/peerj.17282#supplemental-information.

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
