# Peer review of "Fumaric acid production from fermented oil palm empty fruit bunches using fungal isolate K20: a comparison between free and immobilized cells"

_PeerJ, doi:10.7717/peerj.17282_

## Round 0.1 · original submission · Major Revisions

Please address all comments by the reviewers with an emphasis in proper statistical analyses and running additional fermentations.

·

Basic reporting

The overall language use is clear and concise. Point where correction should be made are in the following lines:

50: the phrase “never run out” attributed to the raw material is too strong of a phrase and is not necessarily true. The prior attributes of the material “cheaper, abundant, sustainable, and readily available” already convey that it is a superior material to refined sugar.

172-173: The phrase “oil palm EFB-derived glucose instead of as the substrate” should be rephrased to “oil palm EFB-derived glucose instead of pure refined glucose as the substrate”.

239: the word by is written twice, this needs to be removed.

174: the definition of the acronym VVM is incorrect it should be volume of gas per volume of liquid per minute. Bioreaction Engineering Principles, 3rd edition, John Villadsen.

Regarding the literature referenced in the paper, it is clear that the authors have done a search of the literature, however there were a few points where references were lacking and or more research is needed. The following lines need attention:

221-223: The statement “Immobilization techniques have been widely explored to increase the possibility of a continuous fermentation process, as well as reuse microbial cells for several cycles on an industrial scale.” although, I know this to be true no referencing has been given to support the claim. Please see: Naude, A., & Nicol, W. (2017). Fumaric acid fermentation with immobilised Rhizopus oryzae: Quantifying time-dependent variations in catabolic flux. Process Biochemistry, 56, 8–20. https://doi.org/10.1016/j.procbio.2017.02.027

262-263: I am confused by the reference to fatty acids. Is this a mistake, did you intend to say fumaric acid? And regarding the literature on lignocellulosic materials to fumaric acid this is a decent amount of literature:

Swart, R. M., Brink, H., & Nicol, W. (2022). Rhizopus oryzae for Fumaric Acid Production: Optimising the Use of a Synthetic Lignocellulosic Hydrolysate. Fermentation, 8(6), 278. https://doi.org/10.3390/fermentation8060278

Xu, Q.; Li, S.; Fu, Y.; Tai, C.; Huang, H. Two-stage utilization of corn straw by Rhizopus oryzae for fumaric acid production. Bioresour. Technol. 2010, 101, 6262–6264.

Liu, H.; Hu, H.; Jin, Y.; Yue, X.; Deng, L.; Wang, F.; Tan, T. Co-fermentation of a mixture of glucose and xylose to fumaric acid by Rhizopus arrhizus RH 7-13-9#. Bioresour. Technol. 2017, 233, 30– 33

Liao, W.; Liu, Y.; Frear, C.; Chen, S. Co-production of fumaric acid and chitin from a nitrogen-rich lignocellulosic material—Dairy manure—Using a pelletized filamentous fungus Rhizopus oryzae ATCC 20344. Bioresour. Technol. 2008, 99, 5859–5866.

Deng, F.; Aita, G.M. Fumaric Acid Production by Rhizopus oryzae ATCC® 20344™ from Lignocellulosic Syrup. Bioenergy Res. 2018, 11, 330–340.

Experimental design

The aim and the scope of the research area is well defined. The points that require improvement are in the following lines:

128: Please provide the ATCC or NRRL culture collection numbers in addition to the TISTR numbers for the reference organisms that you have used. This will make your paper more useful to a wider audience and more easily found by search engines.

151-153: No quantification has been given as to the concentration of the sugar in the enzymatic hydrolysate. This is a crucial piece of information. Further no explanation is given to the process of how the sugar solution was treated before being used as the sugar substrate for further fermentations.

173: No information has been given as to how the pH of the medium was controlled at 3.5. Additionally, this pH is lower than reported in literature for any of these organisms, it is not justified and is lower than the mediums reported for either the growth or production steps.

Swart, R. M., Ronoh, D. K., Brink, H., & Nicol, W. (2022). Continuous Production of Fumaric Acid with Immobilised Rhizopus oryzae: The Role of pH and Urea Addition. Catalysts, 12(1), 82. https://doi.org/10.3390/catal12010082

Validity of the findings

The reported findings necessitate a more detailed exposition of the data generated for the experiments outlined in the Materials and Methods section.

189-192: The process explained in the cultivation of the various strains and the final production fermentation is not sufficiently elaborated on. The data in Table 1 does not include any information as to whether these values were obtained from the final time points or an intermediate? No information is given as to whether the experiments were repeated and if so, how many replicas included? Where controls included for any of the conditions? There is no statistical information. Using the mediums reported for the two reference strains the concentrations of 0.12 g/L and 0.9 g/L are not representative of the literature. Concentrations more than 20 g/L should be possible with these strains. This suggests that the reported concentrations may not be reliable for comparison.

Please see: Ronoh, D. K., Swart, R. M., Nicol, W., & Brink, H. (2022). The Effect of pH, Metal Ions, and Insoluble Solids on the Production of Fumarate and Malate by Rhizopus delemar in the Presence of CaCO3. Catalysts, 12(3). https://doi.org/10.3390/catal12030263

229-235: The data that is used in this paragraph (also show in Figure 3 and the supplementary data) does not satisfy a mass balance. Taking one example from the supplementary data condition of 120 g/L initial oil palm EFB-derived glucose, the concentration of glucose decreases to 95.24 g/L after 24 hours, that is a total of 24.76 g/L of glucose consumed. However, over that same period of time 35.39 g/L of fumaric acid was reported to be produced. That is impossible. Either there is a second carbon source present in the medium that has not been accounted for or reported, or the HPLC analysis of the medium has an error. This needs to be rectified across all data sets.

Additional comments

Thank you for the research that you have conducted. Please address the issues highlighted, I believe it will greatly improve your article and make is useful to others furthering the research.

Reviewer 2 ·

Basic reporting

The manuscript entitled “Fumaric acid production from fermented oil palm empty fruit bunches using fungal isolate K20: a comparison between free and immobilized cells” deals with the production of fumaric acid from a new fungal isolate. The manuscript is well-written, however the results are not discussed and the conclusions run hasty. Most specifically:
The title prepares the reader for a comparison between free and immobilized cells. However, in the manuscript the comparison is in no more than 5 lines. A graph showing the strain’s performance in both cases would be really valuable. Moreover, strain immobilization serves as a way to perform multiple batches with the same inoculum. So, I wouldn’t be so fast to just the outcome of this one experiment and try a repeated batch fermentation.
The references used are more than 15 years old! Please try to use more recent publications (5 years old is preferable).
How did the authors monitor sugar consumption and hydrolysis yield?
Line 238: fatty acid or fumaric acid concentration?
Figure 3: Please write “Fumaric acid concentration” instead of yield. Also please use capital L to denote the concentration
Final FA yield was 0.39 g/g. What happened with the rest of the carbon? Did the authors monitor any by-products formation or the fungal growth?

Experimental design

More experiments are required for the work to be completed

Validity of the findings

More experiments are required to validate the outcomes of the work

---

## Round 0.2 · accepted · Accept

Based on the comments by the reviewers, the authors have successfully addressed all their previous concerns; therefore, the manuscript can be accepted.

·

Basic reporting

no comment

Experimental design

no comment

Validity of the findings

no comment

Additional comments

Thank you for incorporating all of the suggestions, your article has improved greatly from the first reading. I appreciate the discussion on your results in line with their possible impact and limit of impact with regard to the techniques used. It is a good article and I hope that it has an impact to further encourage the use of renewable sources of sugar for the production of FA.